# Case Series and Literature Review on Botulinum Toxin Efficacy in Axial Extensor Truncal Dystonia

**DOI:** 10.3390/toxins17080375

**Published:** 2025-07-29

**Authors:** Jarosław Sławek, Iga Alicja Łobińska, Michał Schinwelski, Joanna Kopcewicz-Wiśniewska, Anna Castagna

**Affiliations:** 1Division of Neurological-Psychiatric Nursing, Faculty of Health Sciences, Medical University of Gdańsk, 80-210 Gdańsk, Poland; 2Department of Neurology & Stroke, St. Adalbert Hospital, Copernicus Ltd., 80-462 Gdańsk, Poland; joanna.kopcewicz@gmail.com; 3Medical Faculty, Medical University of Gdańsk, 80-210 Gdańsk, Poland; igalob@gumed.edu.pl; 4Neurocentrum-Miwomed, Neurological Clinic, 80-207 Gdańsk, Poland; szyna777@gmail.com; 5IRCCS Fondazione Don Carlo Gnocchi, Onlus, 20162 Milan, Italy; anna.castagna18@gmail.com

**Keywords:** dystonia, truncal dystonia, axial dystonia, axial extensor dystonia, botulinum neurotoxin

## Abstract

Axial truncal dystonia can present as either flexion or extension, often with a tendency toward lateral movement. Flexion dystonia is more common and may represent a clinical spectrum associated with parkinsonism. In contrast, extensor trunk dystonia is less frequent and exhibits a diverse range of causes. In this paper, we reviewed the literature on axial extensor trunk dystonia. We identified 11 studies involving 49 patients, of which only 10 had idiopathic trunk dystonia. Treatment with botulinum neurotoxin A (BoNT/A) emerged as the most effective therapy; however, many studies did not provide detailed descriptions of the treatment (4/11) and follow-up periods were not specified or short term (up to one–two years). We present four new, well-documented patients with the idiopathic form of extensor trunk dystonia who were treated with BoNT/A with moderate to significant effect according to Global Clinical Impression scale (GCI) and Burke-Fahn-Marsden (BFM) dystonia scale. These cases include long-term follow-up for three patients, all without any adverse events. While the diagnostic process and treatment can be challenging, we recommend using BoNT/A with adjusted doses tailored to the appropriate muscle groups as a first-line treatment.

## 1. Introduction

Truncal dystonia is a hyperkinetic movement disorder characterized by involuntary spasms of the agonistic and antagonistic muscles of the chest, abdomen, and back. This condition leads to abnormal postures of the trunk [1]. Extensor axial dystonia, in particular, is marked by abnormal arching (opisthotonus) and extension of the trunk, resulting from the overactivity of thoracic and/or lumbar paravertebral muscles [2]. This disorder is dynamic, meaning its severity can change based on the patient’s activities and postures [3]. Extensor truncal dystonia may cause pain and tremors, interfere with motor skills, and result in gait dysfunction [1]. Truncal dystonia often suggests an underlying secondary cause, such as exposure to dopamine receptor antagonists or various neurodegenerative and inherited disorders [2]. In cases where the trunk exhibits flexion (with camptocormia being the most severe form) or a lateral flexion (known as Pisa syndrome), the condition may be part of the clinical spectrum of parkinsonism, primarily related to multiple system atrophy (MSA) or Parkinson’s Disease (PD) [4,5]. While truncal flexion is the most common type of trunk dystonia, extensor truncal dystonia is relatively rare, and epidemiological studies on its prevalence have not been reported [6]. Extensor axial dystonia significantly impacts basic life skills and overall quality of life, and managing this type of dystonia presents challenges. Data on effective treatments are limited, with the current standard therapies being dopaminergic and anticholinergic treatments, which show only insufficient efficacy [2]. Due to the rarity of this dystonia presentation there were no formal studies, so the majority of observations are based on case reports and case series. Other treatment modalities, such as physiotherapy, deep brain stimulation (DBS), and Botulinum neurotoxin (BoNT) [1,2,3,4,6], have produced varying results that are often based on individual case reports. Among the available treatments, BoNT appears to be one of the most effective options. Types A and B of BoNT have been formally approved for treating blepharospasm, cervical dystonia, and spasticity, with strong evidence supporting their efficacy [7]. Although truncal dystonia is rare and no formal randomized studies have been published, BoNT has been shown to relieve pain and improve dystonic postures [1,2,3,4,6].

This study aims to review the literature regarding extensor truncal dystonia, specifically analyzing the treatment responses to BoNT injections. Additionally, it presents a new series of video-illustrated case studies demonstrating successful outcomes and long-term follow-ups.

## 2. Results

After a thorough review of the literature, we identified 11 eligible papers, reporting a total of 49 patients with extensor truncal dystonia. Most of these cases were single ones derived from heterogeneous case series of segmental or generalized dystonia. Among the 49 cases, we found 7 children exhibiting opisthotonus as part of the clinical spectrum, with 4 cases associated with inherited disorders. The etiologies varied, comprising tardive generalized dystonia (6 cases), adult-onset X-linked dystonia-parkinsonism (XDP) (7 cases), idiopathic parkinsonism (2 cases), L-dopa responsive parkinsonism (9 cases), Pantothenate kinase-associated neurodegeneration (PKAN) (2 cases), post-infectious (1 case), perinatal (1 case), and gene mutations of uncertain significance in various genes. This included hypomyelinating leukodystrophy-14, NBIA/PKAN, CASK-related disorders, presumed PANK2 mutations (6 cases), and other idiopathic focal dystonia that spread out to paraspinal muscles (1 case). Only 10 patients had an identified etiology categorized as idiopathic; the remainder fell within the spectrum of other disorders or were described as tardive. Additionally, one paper, which included 5 patients, did not specify the etiology at all [1]. All patients received injections into the paraspinal muscles, with occasional injections in the splenius, trapezius, and scalene muscles. Due to the lack of a uniform protocol for BoNT administration and the absence of clear data regarding recommended doses and the number of injection sites, it was not possible to calculate the mean dose, cumulative dose, or the number of neurotoxin injections across all studies reviewed. However, the cumulative dosage for abobotulinumtoxinA (DYSPORT^®^, Ipsen Pharma, Paris, France) ranged from 100 to 1200 units, depending on the muscle groups injected, with a median of 600 units (±288.98 units). For onabotulinumtoxinA (BOTOX^®^, AbbVie, formerly Allergan, Chicago, IL, US), the cumulative dosage ranged from 150 to 700 units, with a median of 295.7 units (±232.45 units). In the majority of the papers, authors reported moderate to good results. However, not all reports were comprehensive; some lacked specific information about the preparation of BoNT/A and doses. Almost all papers reported an improvement in patients who underwent BoNT therapy (10 out of 11 articles), indicating that 46 out of 49 patients experienced any level of improvement. The most significant improvement observed was in dystonia-induced pain. Additional reported improvements included enhanced gait, breathing, patterns of EMG activity, and increased mobility, as related to the specific study goals. The direct comparison of treatment effects was difficult as the treatment goals were differently defined (EMG activity, breathing or pain). In most cases, extensor truncal dystonia was dominant, but was part of a broader involvement of adjacent muscle groups (segmental dystonia). A detailed summary of the analyzed papers, including the number of patients, injected muscles, BoNT preparations and doses, clinical effects, adverse events, clinical outcome measures, and dystonia etiology, is presented in Table 1.

## 3. New Case Reports

In this report, we present 4 new cases of idiopathic truncal extensor dystonia that demonstrated successful outcomes (with long-term follow up in 3) following treatment with BoNT (AbobotulinumtoxinA in 3 and OnabotulinumtoxinA in 1). They represent the group of all patients with this type of dystonia found in our data bases. All patients underwent detailed diagnostic procedures, including genetic testing (in 3), and evaluations for autoimmune and symptomatic etiologies (in all). Video recordings document the patients before and after (4–8 weeks) BoNT/A injections. Treatment effect was assessed by physician according to Burke-Fahn-Marsden (BFM) rating scale and Global Clinical Impression scale (GCI, rated: −3 significant, −2 moderate, −1 mild deterioration, 0—no effect, +1 mild, +2-moderate, +3 significant improvement) pre- and post-injection.

### 3.1. Case No. 1

A 45-year-old male presented with involuntary movements affecting the neck and paraspinal muscles, resulting in pronounced retrocollis to the right side, along with axial extensor and lateral truncal flexion to the right. These movements were exacerbated by walking. His medical history included treatment for arterial hypertension with quinapril and bisoprolol. A family history was negative. MRI of the brain and cervical and thoracic spine, as well as cerebrospinal fluid examination, were all normal. Wilson’s disease was ruled out with normal serum and 24-h urine copper excretion and ceruloplasmin levels. A psychological examination revealed some abnormalities in visuo-spatial function, but overall was normal. Genetic testing for DYT-TOR1A and DYT-THAP1 showed no abnormalities. He was also tested for Stiff-Person syndrome, but anti-GAD antibodies were absent. Finally, he was diagnosed as idiopathic cervical and axial dystonia. L-Dopa treatment proved ineffective. The patient was given biperiden at 4 mg/day, which was increased to 6 mg/day with only mild improvement, which he maintained throughout the observation period. He was then referred to a Botulinum toxin outpatient clinic where he began treatment with injections. The initial treatment with OnabotulinumtoxinA (a total dose of 170 units) targeted both trapezius, latissimus dorsi, and right-side paravertebral muscles, resulting in approximately 40% subjective improvement (mild). Subsequent treatments switched to AbobotulinumtoxinA, with doses increased from 200 units (ineffective) to 600 units in the paraspinal muscles, still only achieving a mild 30% improvement. All injections were performed without ultrasound or electromyography control. In the second year of treatment, the dystonia pattern changed. While cervical dystonia was less severe, the patient began to experience additionally to trunk dystonia dystonic spasms in the left lower limb (with knee flexion) involving the biceps femoris accompanied by severe pain (Appendix A). According to BFM the total score was 34 and GCI was assessed −3). He received AbobotulinumtoxinA injections totaling 1000 units, divided among paraspinal muscles (600 units), the right trapezius (200 units), and the left biceps femoris (initially 200 units, increased to 300 units in subsequent sessions), resulting in a very good effect both in terms of posture and pain (Appendix A). Improvement assessed at BFM scale was 4 and GCI: +2. This positive response lasted for about two months before gradually diminishing, requiring reinjection. He received a total of 18 injections with the same dosage (distribution slightly adjusted to 800 units for paraspinals and 200 units for the biceps femoris) guided by ultrasound in the last sessions, achieving very good clinical effects without adverse reactions. However, at his last visit, he was diagnosed with sigmoid colon cancer and did not return to the BoNT therapy program.

### 3.2. Case No. 2

A 64-year-old male presented with a medical history that included arterial hypertension, hypercholesterolemia, and a right-hand mixed tremor that was non-responsive to L-dopa, as well as low back pain following two discoidectomies (performed 18 and 1 year earlier), preceding the onset of his dystonia. He was admitted due to painful neck and trunk spasms that caused laterocaput to the right and extensor truncal dystonia on the right side (Appendix A). He exhibited a sensory trick, with improvement in his abnormal posture when leaning against a wall or wearing a backpack (Appendix A). Clinical assessment according to BFM scale was 28 and CGI: −3. MRI of the brain and spinal cord (cervical and thoracic) showed only moderate white matter hyperintensities. Cerebrospinal fluid, blood ceruloplasmin, and genetic testing for DYT-TOR1A and DYT-THAP1 were normal, as did anti-GAD antibodies for Stiff-Person Syndrome. Oral treatments with L-Dopa, biperiden, and baclofen were ineffective. Therefore, he was diagnosed with idiopathic cervical and truncal dystonia. BoNT injections were initiated in the trapezius and paraspinal muscles (longissimus thoracis, iliocostalis lumborum, spinalis thoracis) on the right side. The initial clinical effect was mild after the first session (AbobotulinumtoxinA: 500 units, with 200 units to the trapezius and 300 units to the paraspinal muscles). However, improvement increased in the following months with doses reaching 750 units and finally 1000 units, including injections into the right splenius capitis (150 units), trapezius (150 units), and levator scapulae (150 units), as well as 600 units to the paraspinal muscles, resulting in very good clinical outcomes both in posture and pain (Appendix A). The BFM score was assessed as 1, and CGI as +3. There were no reported adverse events, and the clinical effect gradually wore off between the third and fourth months following each injection. He has undergone 3–4 injections per year for the past 12 years, consistently achieving good results with a reduced total dose now down to 600 units (400 units for paraspinal muscles and 200 units for trapezius on the right side), with injections guided by ultrasound.

In Cases 1 and 2, the vial containing 300 units of AbobotulinumtoxinA was diluted with 1.5ml of normal saline, and 500 u vials with 2.5 mL.

### 3.3. Case No. 3

A 47-year-old female with no comorbidity experienced the onset of mild lower limb dystonia at the age of 6, followed by writer’s cramp at 13, and cervical and extensor trunk dystonia at 42. She has a positive family history, with one brother affected by hemidystonia involving the upper and lower limbs on the left side. Clinically, she initially presented with laterocaput to the right and trunk extension to the left. Her gait was disturbed by abnormal posture (Appendix A). The clinical assessment according to BFM rating scale was scored as 29 and CGI as −3. She never complained about pain. Due to her positive family history and the early onset of dystonia, she underwent genetic testing, including DYT-TOR1A and next-generation sequencing (NGS) dystonia panels, with unremarkable results. She was then referred to the Botulinum toxin clinic, where she received injections of AbobotulinumtoxinA under ultrasound guidance six times over the past two years, with good effects and no adverse events (Appendix A). Marked improvement was noticed at BFM scale—rated 10 post-injection and CGI was +2. The muscles involved in the trunk extension were injected: 125 units to the right longissimus dorsi at the D8-D9 level, and 150 units to the left side. For the laterocaput, the injected muscles included: levator scapulae (100 units), splenius capitis (100 units), longissimus capitis (75 units), and scalenes (75 units).

One vial of 500 units of AbobotulinumtoxinA was diluted in 2 mL of normal saline.

### 3.4. Case No. 4

A 44-year-old female has had hearing loss since birth and no other comorbidity, along with dystonia for the past 10 years. At onset, she experienced involuntary elevation of her right shoulder. Three years before admission, she began to exhibit extensor trunk dystonia, characterized by trunk bending backward and to the left, accompanied by left hip elevation and slight internal rotation of the left lower limb. The dystonic movements worsened during walking, but significantly improved while lying down (sensory trick) (Appendix A). BFM was scored 18 and CGI: −2. She reported experiencing severe back pain during walking and daily activities. There was no family history of dystonia. Wilson’s disease and Stiff Person Syndrome were ruled out, with no anti-GAD antibodies detected. Brain and spine MRI results were normal. As of now, no genetic testing has been performed. She was treated with anticholinergics (pridinol) and L-Dopa, but showed no significant improvement. Prior to her referral, she had been treated with botulinum toxin injections elsewhere, targeting only cervical muscles, with no effect. As a new patient, she has been treated at our center only once so far. Multiple injections were administered under ultrasound guidance to the latissimus dorsi, spinalis thoracis, longissimus thoracis, iliocostalis thoracis and lumborum on the left, and levator scapulae on the right, using a total dose of 300 units of OnabotulinumtoxinA. During her control visit after four weeks, improvement noticed after just one week and no adverse events were reported. Her abnormal posture and gait improved, and her pain was reduced, leading to a substantial enhancement in her daily activities, which she rated at 60% improvement (good effect) (Appendix A). According to BFM scale she improved and scored 9 and CGI: +2.

OnabotulinumtoxinA vial of 100u was diluted in 2 mL of normal saline.

## 4. Discussion

Truncal extensor dystonia presents a significant diagnostic and therapeutic challenge. In the reviewed literature, only a minority of patients are diagnosed with idiopathic dystonia, 10 out of 49 in the 11 analyzed papers. When making a clinical diagnosis, it is crucial to account for the wide spectrum of possible etiologies, including inherited forms of dystonia that affect the neck and trunk, parkinsonism, tardive dystonia, and Stiff-Person Syndrome, as these conditions may require specific treatments.

Even in cases of inherited or tardive syndromes, Botulinum toxin (BoNT) local injections can be beneficial as symptomatic treatment, although this is considered off-label therapy. Similarly, other published treatments using anticholinergics, baclofen, tetrabenazine, or clozapine often show only partial or no effectiveness. Local BoNT injections into the affected muscles, akin to procedures used for cervical or limb dystonia, appear to be the preferred treatment. When adhering to the maximum allowed in a summary of product characteristics dosages, this procedure is also considered safe.

For refractory cases, bilateral DBS of the internal globus pallidus (GPi) may be an option, with expectations of favorable outcomes as seen in segmental or generalized inherited and tardive dystonia [16,17,18]. Both medication and surgical procedures come with potential adverse events. Local treatment with the use of BoNT, which usually involves repeat injections every 3 to 4 months, is a more comfortable and safe therapy option.

A review of the literature shows that many authors reported moderate to good results in their cases. However, the majority of studies discuss patients who received only a single injection, while some involved repeated injections over two years. Our 3 new cases (no 1–3), with follow-up of 6, 12 and 2 years respectively, demonstrated sustained positive effects and safety profiles, indicating that BoNT can be an effective long-term treatment. Notably, pain disappeared in all the patients complaining about it, following repeated injections. Evidence from a recent study on cervical dystonia showed that 48.1% of patients experienced at least a 30% reduction in pain, with 34.4% achieving a ≥50% reduction from their baseline. Furthermore, 10.3% of these patients became pain-free, and pain relief was sustained over five injection cycles, with trends showing incremental improvements with each successive cycle [19].

The effectiveness of BoNT injections is linked to accurate diagnosis, proper identification of the involved muscles, effective needle placement, and appropriate adjustment of the BoNT dosage. Due to truncal extensor dystonia rare nature, detailed anatomical landmarks are limited in the existing literature. In our patient group, we adapted a treatment protocol originally recommended for Stiff-Person Syndrome. The muscles responsible for spinal extension include the spinalis thoracis (medial), longissimus thoracis (middle), and iliocostalis lumborum (lateral), which collectively form the erector spinae in the upper lumbar region, with attachments to the cervical and thoracic vertebrae as well as the sacrum and iliac bones. Other relevant trunk muscles that assist with trunk extension, stabilization, and rotation include the multifidus and quadratus lumborum [20]. Distinguishing these large and lengthy muscles can be challenging, so it’s advised that injectors palpate the entire muscle group and administer multiple injections along one side of the spine. The Yale protocol is commonly utilized for this purpose [20]. The spinal erectors are generally easily palpable, and we recommend performing injections while the patient is seated. Other cervical muscles can be identified based on the Col-Cap concept, reflecting specific head and neck postures [21]. We present a schematic illustration of the back muscles involved in trunk positioning, using color coding for different muscle layers identified through ultrasound examination (Figure 1). For larger patients with thinner skin and subcutaneous tissue, or when targeting deeper muscles, reaching them may require guidance. We suggest employing ultrasound-guided (US) injections, as there is accumulating evidence that this approach is more effective [22,23,24,25,26]. Eventually, EMG can be added as a second guide. In all reported new cases EMG was used, but only at the initial phase to make a proper diagnosis (it may be helpful to detect the specific continuous activity typical for e.g., Stiff Person Syndrome), but was used in one case in the muscle selection process but not to monitor injections or treatment effect. Additionally, dosages should be tailored to each individual based on their weight and muscle size, but we recommend starting with lower doses—while ensuring they are not too low, since these are large muscles with numerous attachments—and subsequently escalating doses in future injections. Significant side effects have not been reported in the literature (except one patient who experienced neck extensor weakness [15]) or in our case series, and notable weakness in the injected trunk muscles appears to be rare. Our own experience (4 new cases) showed the effectiveness and safety of this treatment, however, regarding the small number of cases, conclusions should be treated cautiously. We recommend BoNT injections to be considered as the effective treatment for axial extensor dystonia affecting the trunk. However, the list of possible treatment failures is relatively long, starting from wrong diagnosis, evolving diseases (like Wilson’s Disease) if not appropriately treated, wrong identification of dystonia pattern, muscles involved, doses used or lack of injection guidance (US and or EMG) followed by rare causes like long lasting dystonia with rheological changes within muscles (contractures) or BoNT antibodies. All of them may contribute to overall effect of the treatment [20,21]. Therefore, we called this type of dystonia challengeable.

There are also limitations of our study that should be emphasized: the data heterogeneity, lack of Randomized Controlled Trials RCTs, and selection bias (many papers on BoNT use in extensor truncal dystonia were only a part of a large series of phenomenologically and etiologically different disorders with accompanying trunk dystonia). Therefore, to our opinion adding new evidence seems to be especially valuable.

Truncal extensor dystonia is etiologically distinct from flexion dystonia identified as camptocormia in Parkinson’s Disease patients or Pisa Syndrome in Multiple System Atrophy. There is no consensus on the dystonic origin of these flexed postures (dystonia of flexors or weakness of the back muscles due to their myopathic changes) and the treatment results are inconsistent [27].

## 5. Materials and Methods

To conduct a systematic literature search, we utilized the Medline database, along with Google and Google Scholar, focusing our research until December 2024. We employed a range of search terms that provided optimal sensitivity and specificity, including “truncal dystonia,” “axial dystonia,” “extensor truncal dystonia,” “opisthotonos,” “neurotoxin,” and “botulinum toxin.” Additionally, we performed both forward and backward literature searches. The forward search involved identifying articles that cited the article of interest, while the backward search included a manual review of the references in the article being examined. The inclusion criteria for the literature comprised any study that contained information about patients with extensor truncal dystonia who were receiving BoNT therapy. We focused on articles published from 1989 onward, the year when onabotulinumtoxinA (BOTOX^®^) was first approved for medical use.

Extensor truncal dystonia was defined as the involuntary hyperactivation of the chest, abdomen, and back muscles, leading to an extended posture along the sagittal axis with a possible lateralization. Our analysis considered studies in which BoNT injections were both single and/or repeated. We included only original case series, reports, and literature reviews, adhering to the PRISMA guidelines to identify articles for our review (see Figure 2). Only articles published in English and involving human subjects were included. Thus, we presented our own new four cases, detailing injection schedules and demonstrating successful treatment with BoNT in all of them.

Abstracts were screened and full texts were assessed for inclusion by a single investigator. Appropriate studies proceeded to data extraction, which was focused on clinical presentation of extensor truncal dystonia despite its origin and Botulinum toxin intervention. Authors are aware of potential bias, but due to the heterogeneity and lack of standardized tests used to compare the efficacy of Botulinum toxin in selected papers the potential use of e.g., Quadas-2 tool was difficult. Therefore, we do not state it is a systematic review. However included papers involve the etiologically heterogeneous group of patients what may be a confounding factor, we have focused on a specific phenotype of dystonia to review the possible BoNT effectiveness regardless the etiology. It may better show the real-life practice.

## Figures and Tables

**Figure 1 toxins-17-00375-f001:**
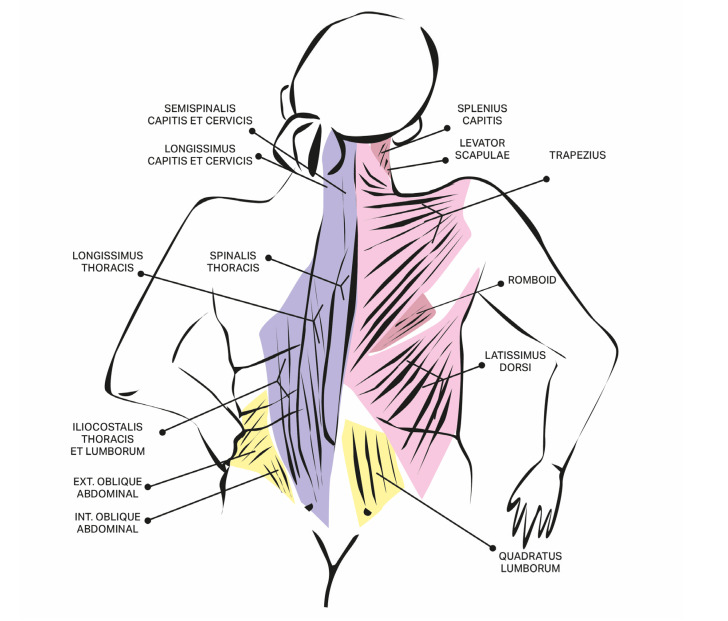
The anatomy of the relevant for the treatment of extensor truncal dystonia muscles along with ultrasound guidance enabling the precise muscle injections.

**Figure 2 toxins-17-00375-f002:**
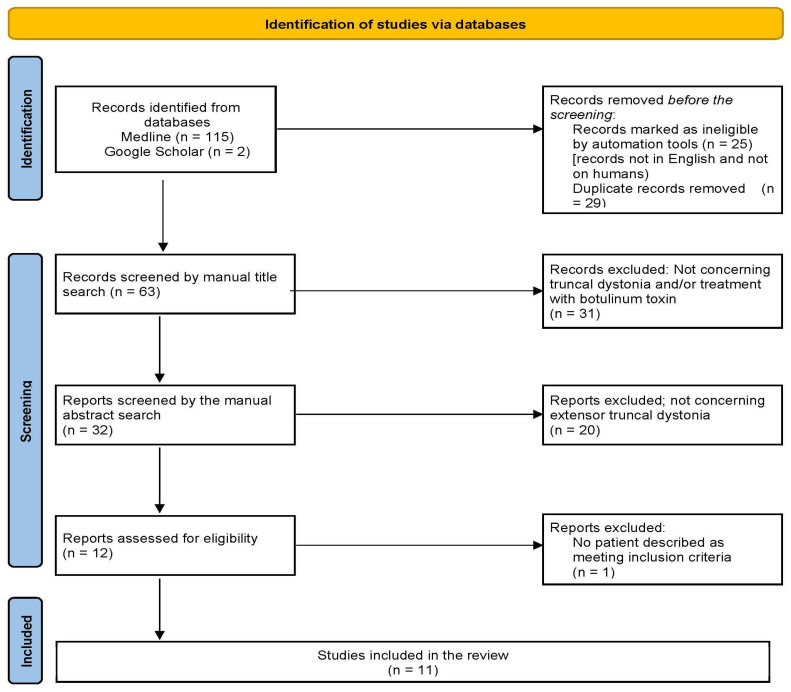
Schematic presentation of the selection process according to the PRISMA guidelines used to identify the articles reviewed in this work.

**Table 1 toxins-17-00375-t001:** Review of extensor truncal dystonia cases reported in the literature with efficacy of Botulinum toxin treatment.

Author, Year, Title	No of Patients with Extensor Trunk Dystonia in the Whole Group	Injected Muscles (as Specified by Authors)	Drug, Dosage [Units]	Clinical Effect (as Specified by Authors)	Adverse Effects	Method of Assessment/Scale	Etiology of Dystonia
**Mezaki, 1994** *Optimisation of botulinum treatment for cervical and axial dystonias: experience with a Japanese type A toxin* [8]	2	Cervical and thoracic paravertebral muscles	**Crystallised type A botulinum toxin** (**CS-BOT; Chiba Serum Institute, Chiba, Japan)** The maximal dose: 150 units/muscle, 300 units/session	1st patient: Regained the ability to walk after eight injections (cumulative dose 2100 units) 2nd patient: Abnormal posture was corrected (cumulative dose 700 units)	No data	Subjective ranged from 0 (no improvement) to 100 (complete recovery), as rated by the patient objective scores (videotape assessed by neurologist)	1st patient: Tardive generalised dystonia for 12 years 2nd patient: Idiopathic axial dystonia for 5 years
**Lagueny, 1995** *Involvement of Respiratory Muscles in Adult-Onset Dystonia: A Clinical and Electrophysiological Study* [9]	1	Splenius and paraspinal muscles	**AbobotulinumtoxinA (Dysport^®^)** Single dose per splenius muscle was 100 units for one injection (3 injections through the year, cumulative dose for both splenius muscles during one year was 600 u). Single dose per paraspinal muscles was 200 u for one injection (3 injections throughout the year, cumulative dose for both paraspinal muscles during one year was 1200 u).	Improvement of axial dystonia and respiratory problems More regular pattern of EMG activity	No data	Clinical general assessment EMG activity	Idiopathic axial dystonia following isolated significant retrocollis
**Quirk, 1996** *Treatment of Nonoccupational Limb and Trunk Dystonia with Botulinum Toxin* [10]	1	T4-T10 paraspinal muscles	**AbobotulinumtoxinA (Dysport^®^)** Paraspinal muscles—750 u	Patients’ assessment: very good Clinical assessment: increased mobility	No data	Patients graded their overall assessment of response to treatment as: no benefit, fair, good, very good, or excellent. Pain assessment: 0 = no improvement and **3** = complete pain relief Improvement in posture assessed clinically by two neurologists and scored on a 5-point scale (where 0 = no change and 4 = normal posture), and functional benefits were recorded descriptively.	Idiopathic
**Comella, et al., 1998** *Extensor Truncal Dystonia: Successful Treatment With Botulinum Toxin Injections* [1]	5	longissimus and spinalis muscle groups The muscles were injected in the lower back from the level of the tenth thoracic vertebrae to the second lumbar vertebrae.	**Onabotulinum toxinA (BOTOX^®^)** Patients with moderate hypertrophy received 150–200 u; patients with marked hypertrophy received of 200–300 u; 2 or 3 sites on each side, with 25–50 u per site;	Objective improvement on videotaped scores (37%); subjective improvement (46%); improvement in pain (30–80%) all patients reported less pain and increased range of motion	no adverse effects	Modified from the Burke-Fahn-Marsden videotape protocol examiner blinded to the treatment order, scale using 0 (no dystonia) to 10 (extreme arching or bending). Patients’ self-assessment: from −100% (worse) to +100% (improved) on position, pain, and mobility	Adult-onset; no data on etiology
**Bonnani, 2007** *Botulinum Toxin Treatment of Lateral Axial Dystonia in Parkinsonism* [11]	9	paraspinal muscles 2 to 2.5 cm lateral to spinous processes at level L2-L5 on the side of the trunk	**AbobotulinumtoxinA (Dysport^®^)** total dose of 500 u	6/9 patients showed improvement in grading of function (TDDS) and pain (VAS); two patients did not benefit The improvement of lateral bending responding to the treatment: 50% to 85.7% Marked improvement of posture. 7/9: a remarkable improvement of pain	No adverse effects	Videotaping by an examiner blind to treatment and assessed with the Trunk Dystonia Disability scale (TDDS). A mobile wall goniometer used to calculate the degrees of trunk inclination; Pain rated with a visual analogue scale (VAS: 0–10)	L-dopa responsive parkinsonism
**Rosales, 2011** *The Broadening Application of Chemodenervation in X-Linked Dystonia-Parkinsonism (Part II): An Open-Label Experience With Botulinum Toxin-A (Dysport) Injections for Oromandibular, Lingual, and Truncal-Axial Dystonias* [12]	7	erector spinae, bilaterally	**AbobotulinumtoxinA (Dysport^®^)** Median total dose: 750 u (500–1000 u)	Global Dystonia rating scale median (range): 2 (1–3) Pain VAS reduction at week 4—median (range): 50% (25–75) in 10 out of 12 cases with pain	No adverse effects	The global dystonia rating scale (DRS; ranging from 0 = no effect to 4 = complete dystonia abolition) visual analog scale (VAS: 0–100%)	Adult-onset X-linked Dystonia-Parkinsonism (XDP)
**Voos, 2013** *Case Report: Physical therapy management of axial dystonia* [13]	1	right superior and medium trapezius, scalene and erector spinae;	**Botulinum toxin type A (preparation not specified)** 1st year: 400 u four times a year After one year of treatment: 300 u four times a year	pain relief, no functional improvement	No data	Parts I, II and III of Toronto Western Spasmodic Torticollis Rating Scale (TWSTRS-I, TWSTRS-II and TWSTRS-III), Berg Balance Scale (BBS), Six-Minute Walk Test (6-MWT), and the motor domain of Functional Independence Measure (FIM-motor) before and after the two-year treatment and after the one-year follow-up	Idiopathic
**Ehrlich, 2016** *The phenomenology and treatment of idiopathic adult-onset truncal dystonia: a retrospective review* [6]	2	injected different muscle groups -not specified	**OnabotulinumtoxinA (BOTOX^®^)** botulinum toxin injections were not standardized, different injectors employed different techniques; a total dose of 700 u per patient	No improvement	No data	Videotapes evaluated by senior movement disorder physicians in the outpatient clinic of an urban academic center	Idiopathic
**Mehta, Lal, 2019** *Neurodegeneration with Brain Iron Accumulation: Two Additional* *cases with Dystonic Opisthotonus* [14]	2	paraspinal muscles	**Abobotulinum toxinA (DYSPORT^®^)**; Patient—500 u Periodic botulinum toxin therapy—600 u per session	1st patient: Poor response to the treatment 2nd patient: Regained ability to walk	No data	Patient—no data Video analysis—no detailed data on assessment method	1st patient: pathogenic variant in Exon 1 and 3 of PANK2 gene 2nd patient: compound heterozygous gene mutations of uncertain significance in PANK2 gene
**Mehta et al., 2020** *Spectrum of Truncal Dystonia and Response to Treatment: A Retrospective Analysis* [2]	12	Paraspinal muscles	**AbobotulinumtoxinA (DYSPORT^®^)**; Unilateral → median (SD, range) dose 143.33 u ± 54.32 (100–250 u) Bilateral → median dose 286.67 u ± 108.65 (200–500 u)	Average subjective response of improvement was 30.8 ± 21.8% (0–70%)	No adverse effects	Single question evaluating improvement in pain, dystonia and functional status scored 0–100 Improvement assessed only for truncal dystonia in patients with multifocal, segmental, or generalized distribution Response measured only for the first injection in patients who underwent multiple sessions	2 cases of Parkinson’s Disease, 4 tardive dystonia, 4 idiopathic, 2 *PKAN* (Pantothenate kinase-associated neurodegeneration)
**Hull, 2021** *Botulinum Neurotoxin Injections in Childhood Opisthotonus* [15]	7	paraspinal muscles	**Onabotulinumtoxin A (BOTOX^®^)**; Paraspinals—between 120 to 300 u (average 215.7 u)/divided into three to five injections on each side. Total dose administered 150–650 u (average 341.4 u) or 16.7 to 23.8 u/kg (average 19.6 u/kg).	All patients showed an improvement in opisthotonus within 3–14 days (average 6.1 days).	no adverse effects; except for one: neck extensor weakness, resolved over a period of weeks with no recurrence	Improvement was assessed on the basis on “to be crossed out examination of each child, and improvement deemed to be of a meaningful and sufficient degree by caregivers of all injected children”	3 acquired (Post-infectious, Perinatal HIE, Tardive dystonia), 4 genetic (Hypomyelinating leukodystrophy-14, NBIA/PKAN, *CASK*-related disorder, presumed)

## Data Availability

The original contributions presented in this study are included in the article. Further inquiries can be directed to the corresponding author.

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
