# Peer review of "Case Series and Literature Review on Botulinum Toxin Efficacy in Axial Extensor Truncal Dystonia"

_toxins, 2025, doi:10.3390/toxins17080375_

Round 1

Reviewer 1 Report (Previous Reviewer 1)

Comments and Suggestions for Authors

Dear authors, 

I have thoroughly reread your article. Although I am not fully convinced that extensor-type truncal dystonia will survive the long-term nosological discussions, I would now argue that it merits a more careful appraoch - as outlined in your paper that I now support for publication.

Author Response

Thank you very much for taking the time to review this manuscript.

Reviewer 2 Report (Previous Reviewer 2)

Comments and Suggestions for Authors The authors have adequately addressed all comments, which are clearly reflected in the revised text.

Author Response

Thank you very much for taking the time to review this manuscript.

Reviewer 3 Report (Previous Reviewer 3)

Comments and Suggestions for Authors

In this paper, the authors present a literature review on axial extensor dystonia of the trunk and present new cases of patients treated with Botulinum Toxin for this condition. This topic is interesting and little discussed in the current literature. The subject is therefore relevant.

The authors properly addressed all of my comments and suggestions, and I am satisfied with the improved version of the manuscript. I want to applaude the authors for their good work.

Author Response

Thank you very much for taking the time to review this manuscript.

Reviewer 4 Report (New Reviewer)

Comments and Suggestions for Authors

Title

Too long and descriptive. The use of "New Cases with a Good Response" is vague and not academically rigorous. It should be more precise (e.g., “Case Series and Literature Review on Botulinum Toxin Efficacy in Axial Extensor Truncal Dystonia”).

Abstract 

  • The mention of "successful" treatment should be backed by objective measures even in the abstract.

  • The phrase “many studies did not provide detailed descriptions” is vague and could be more precise.

Introduction

  • Some sentences are repetitive (e.g., the role of BoNT is mentioned multiple times).
  • The literature gaps are mentioned but not sharply defined—what specifically is missing? Dosing data? Outcome metrics? Standardization?

Materials and Methods

  • The search strategy lacks detail: databases other than Medline and search terms used should be better specified.
  • There is no mention of how studies were screened (e.g., number of reviewers, risk of bias assessment).
  • Inclusion of Google and Google Scholar without clarification raises concerns about reproducibility.

Results

  • Lack of standardized outcome measures across studies is acknowledged but could be more analytically addressed (e.g., via a narrative synthesis or score range comparison).
  • Results could be split more cleanly between literature review and new case reports.
  • The heterogeneity of etiologies (e.g., XDP, PKAN, post-infectious) is profound and undermines any collective interpretation. Pooling these studies without stratification makes the conclusions unreliable.

New Case Reports

  • Some case descriptions are overly detailed for a peer-reviewed article (e.g., specific MRI results and minor comorbidities).
  • Lack of consistency in reporting structure (e.g., not all cases mention follow-up duration or adverse event monitoring uniformly).
  • Case 4 had only a single injection, which weakens the strength of its inclusion in a “long-term” efficacy context.

Discussion

  • The discussion sometimes drifts into educational commentary (e.g., anatomy of spinal extensors) rather than synthesis of results.
  • Statements such as "our experience showed how effective and safe that treatment is" are overly definitive given the small sample.
  • More critical reflection on the unsuccessful cases from the literature would be useful.

Figures and Tables

  • Table 1 is dense and hard to read due to text-heavy content.
  • Figure captions could be more informative.
  • Consider splitting Table 1 by aetiology (idiopathic vs. secondary).
  • Annotate Figure 2 to show specific injection targets more clearly.

This topic is worth publishing, but only after these substantial deficiencies are addressed.

Comments on the Quality of English Language

Acceptable 

Round 2

Reviewer 4 Report (New Reviewer)

Comments and Suggestions for Authors

The authors have made substantial efforts to address the reviewer’s comments . They appropriately clarified vague language, restructured content for clarity.

However, certain areas remain only partially resolved. The Materials and Methods section still lacks critical methodological transparency—such as a clear description of the number of reviewers involved in screening, any risk of bias assessment, or detailed reproducibility of the search strategy.

These omissions limit the rigor of the literature review component. Furthermore, while the authors justify the inclusion of heterogeneous case types based on symptomatic treatment goals, this approach risks oversimplifying the clinical variability and may confound interpretation of BoNT efficacy.

Author Response

This manuscript is a resubmission of an earlier submission. The following is a list of the peer review reports and author responses from that submission.

Round 1

Reviewer 1 Report

Comments and Suggestions for Authors

Dear authors,

thank you for giving me the opportunity to review your paper on Axial Extensor Dystonia of the Trunk.

Although I found the topic of your paper worthy of exploration, there are numerous and to my opinion relevant issues that keep me from suggesting publication of the paper in TOXINS in the current form. Among them ...

The authors should first decide whether they want to do a review or a case series. I do not see value in mixing these two reporting styles.

Second, only looking at extensor type dystonia is like only looking at retrocollis when talking about cervical dystonia. When we write a review we need to find constructs that advance our understanding of the disease and not delve into certain types of phenomenology.

The graphic is not adequate for a scientific journal. Pls find a better and more recognizable way to make your point, that can be taken for actual anatomical structures.

The ultrasound graphs are far too small, making the whole idea to demonstrate US pictures not useful.

I do hope you can use my advice to make this a better scientific publication.

Reviewer 2 Report

Comments and Suggestions for Authors

see attached file please 

Comments on the Quality of English Language

need minor revision

Reviewer 3 Report

Comments and Suggestions for Authors

In this paper, the authors present a literature review on axial extensor dystonia of the trunk and present new cases of patients treated with Botulinum Toxin for this condition. This topic is interesting and little discussed in the current literature. The subject is therefore relevant.

Here are a few comments to improve their manuscript.

  • Table 1:
    • Please present the information more concisely. Exhaustive information is important, but try to summarise a bit more and use point form instead of full sentences. This would make the data easier to consult. 
    • In the row on Comella et al.'s 1998 paper, "The muscles were injected in the lower back from the level of the tenth thoracic vertebrae to the second lumbar vertebrae" should appear in the third column instead of the fourth.
  • Case Reports: 
    • Please explain how those cases were chosen, and if data was collected retrospectively or prospectively. Did you have other cases that you decided not to present because of bad outcomes?
  • Discussion:
    • Please compare your results to what is known about flexion dystonia treatment with botulinum toxin in the literature.
    • Please explore the limitations of your review and case reports.
    • Please discuss the need for future research on axial extensor dystonia of the trunk.
  • Figure 2:
    • The first legend about layers of muscles is confusing. It's not clear if you refer to the total number of layers that should be found, following the colour code, or to the layer at which the identified muscle should be found. Leaving the drawing and the ultrasound images might be enough and less confusing.
    • Please identify bony landmarks on ultrasound images.